# LaCo: Efficient Layer-wise Compression of Visual Tokens for Multimodal Large Language Models

## Abstract

Existing visual token compression methods for Multimodal Large Language Models (MLLMs) predominantly operate as post-encoder modules, limiting efficiency. To address this limitation, we propose LaCo (Layer-wise Visual Token Compression), a novel framework for effective token compression within the vision encoder's intermediate layers. LaCo introduces two core components: 1) a layer-wise pixel-shuffle mechanism that systematically merges adjacent tokens through space-to-channel transformations, and 2) a residual learning architecture with non-parametric shortcuts that preserves critical visual information during compression. Extensive experiments indicate that LaCo outperforms all existing methods when compressing tokens in the vision encoder's intermediate layers, demonstrating superior effectiveness. In addition, compared to external compression, our method improves training efficiency beyond 20% and inference throughput over 15% while maintaining strong performance.

## 1 Introduction

Multimodal Large Language Models (MLLMs), like LLaVA (Li et al., 2024a; Liu et al., 2023a; 2024; 2023b), InternVL (Chen et al., 2024e;d), and QwenVL (Bai et al., 2023; Wang et al., 2024b), have demonstrated impressive capabilities in fusing visual and linguistic modalities. These models significantly enhance performance across a wide range of vision-language tasks, including visual question answering, image captioning, and beyond.

Typically, most MLLMs follow the LLaVA framework, in which visual inputs are first encoded into embeddings using a visual encoder and subsequently transformed into a representation compatible with the large language model (LLM) through a projector. The visual embeddings are then concatenated with textual inputs and processed jointly by the LLM. In practice, the number of visual tokens can vary from hundreds to thousands, significantly increasing the running time of MLLMs. Recent studies (Shang et al., 2024; Ye et al., 2024; Chen et al., 2024b) have shown that the visual information often contains a considerable degree of redundancy. Therefore, visual token compression has emerged as a growing area of interest in multimodal learning research.

Previous approaches to visual token compression in multimodal models have mainly focused on external mechanisms applied after the visual encoder. For instance, InternVL utilizes pixel-shuffle (Shi et al., 2016) to merge adjacent tokens within a grid, while QwenVL applies a two-layer MLP to aggregate the neighboring tokens following the 32-layer ViT encoder. Gemma3 (Kamath et al., 2025) adopts the average pooling technique to reduce the number of tokens generated by the SigLIP (Zhai et al., 2023) encoder. However, these methods do not fully exploit potential efficiency gains in visual token compression because they neglect the opportunities within the encoder's intermediate layers.

In this study, we propose **LaCo**, a **La**yer-wise token **Co**mpression method that incorporates the pixel-shuffle and residual connection to effectively prune visual tokens. Specifically, our approach inserts a Patch Merge Layer (PML) after the $k$-th layer of the vision encoder to perform token compression. The PML utilizes pixel-shuffle to merge adjacent visual tokens and then employs a two-layer MLP to map the dimensionality of the merged tokens to the original size. In addition, to mitigate the information loss problem when compressing tokens, inspired by Chen et al. (2024a), we introduce an extra non-parametric shortcut to the PML to let the model learn residuals based on

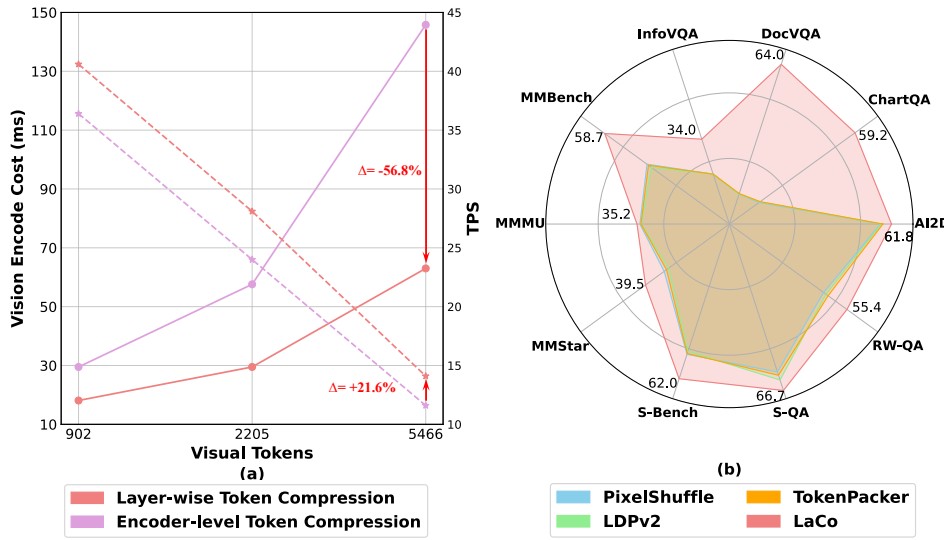

Figure 1: (a) The solid and the dotted represent vision encoding time (VET) and tokens per second (TPS), respectively. Layer-wise token compression achieves a 56.8% reduction in VET and a 21.6% increase in TPS. (b) LaCo outperforms other compression methods. All methods are implemented on AIMv2.

the space-to-channel operation. The proposed PML with residual connection effectively compresses tokens with minimal information loss.

We evaluate the effectiveness of our methods through extensive experiments, which are conducted following the guidance outlined by Li et al. (2024a). Our LaCo outperforms all existing methods when compressing tokens within the intermediate layers of the vision encoder across all scenarios. Compared to external compression, inner-layer compression with LaCo improves training and inference efficiency by over 20% and 15%, respectively, while maintaining strong performance. Furthermore, we conduct a comprehensive exploration of layer-wise token compression by placing LaCo at different depths of the encoder and applying vision encoders including AIMv2 (Fini et al., 2024), SigLIP (Zhai et al., 2023), and InternViT (Chen et al., 2024e), demonstrating the flexibility and generalizability of LaCo.

Our contributions are summarized as follows:

- We propose **LaCo**, a layer-wise visual token compression method that integrates pixel-shuffle and residual connections, significantly improving the training and inference efficiency of MLLMs with only a minor performance drop.

- We conduct extensive experiments with various compression methods, where LaCo demonstrates superior performance and clearly stands out.

- We perform in-depth analyses by inserting LaCo at various depths of the encoder and applying different vision encoders. This exploration not only further validates the effectiveness of our method but also identifies the optimal layer for internal token compression.

## 2 RELATED WORKS

### 2.1 MULTIMODAL LARGE LANGUAGE MODELS (MLLMS)

Recent advancements in MLLMs have primarily focused on the integration of LLMs with advanced visual encoders. This integration is typically achieved through a lightweight projector, facilitating the alignment of vision and language features. For instance, models such as BLIP (Li et al., 2023b; Dai et al., 2023), MiniGPT4 (Zhu et al., 2023), and QwenVL (Bai et al., 2023) utilize the Q-Former (a cross-attention mechanism) to align and distill informative visual features. Flamingo (Alayrac

et al., 2022)introduces gated cross-attention layers to inject encoded visual information into the language model, thereby enhancing multimodal reasoning capabilities. The LLaVA series (Li et al., 2024a; Liu et al., 2023a; 2024; 2023b) adopts a two-layer MLP structure to bridge vision and language models, a design that has influenced subsequent models involving VILA (Lin et al., 2023), ShareGPT4V (Chen et al., 2023), and CogVLM (Wang et al., 2023). These developments highlight the rapid evolution of MLLMs, significantly advancing their capability to handle complex multimodal tasks (Lan et al., 2025).

## 2.2 VISUAL TOKEN COMPRESSION

Visual token compression techniques have been proposed to reduce the inference complexity for MLLMs. Mainstream models, such as Qwen (Wang et al., 2024b), InternVL (Chen et al., 2024d), and Gemma3 (Kamath et al., 2025), utilize two-layer MLP models, pixel-shuffle operations, and 2D average pooling operations, respectively, to achieve visual token compression. In addition to these general approaches, several specialized methods have been designed to enhance the effectiveness of visual token compression (Lan et al., 2024). TokenPacker (Li et al., 2024e) downsamples visual tokens to generate queries and then employs point-to-region attention to extract information from the original tokens, thereby preserving fine-grained details in the compressed tokens. LDP (Chu et al., 2023; 2024) and Abstractor (Cha et al., 2023) use multi-layer convolutional structures to facilitate local token interactions while maintaining spatial relationships. Despite their advantages, these methods perform compression outside the vision encoder. In contrast, our approach introduces compression within the encoder itself, enabling more efficient feature learning. Moreover, we incorporate a residual connection mechanism to mitigate information loss during compression, thereby achieving better performance with reduced computational overhead.

## 3 METHODOLOGY

In this section, we first briefly introduce MLLMs and formulate the visual token compression problem. Afterwards, the patch merge layer with residual skip connection is proposed to mitigate the information loss problem during compression.

### 3.1 MULTIMODAL LARGE LANGUAGE MODELS (MLLMS)

MLLMs, typically exemplified by LLaVA (Li et al., 2024a) encompass a visual encoder, a projector, and an LLM. MLLMs process a pair of visual and textual inputs, denoted as $(T, V)$, where $T$ is the textual input and $V$ is the visual input. The visual encoder extracts features from $V$ and converts them into $N$ visual tokens $E_v = \{v_1, v_2, .., v_N\}$. These tokens are subsequently projected into the textual embedding space. The textual inputs $T$ are mapped to $M$ text tokens $E_t = \{t_1, t_2, ..., t_M\}$ via the text encoder, generally $N \gg M$. The visual and text embeddings are concatenated and used as input to the LLM, which performs autoregressive generation to produce the output sequence.

### 3.2 VISUAL TOKEN COMPRESSION

Since the visual tokens have a high level of redundancy (Shang et al., 2024; Ye et al., 2024; Chen et al., 2024b), compressing visual tokens is necessary for reducing the memory usage and improving the training and inference efficiency. Formally, given a set of visual tokens $E_v$, a compression module $C$ is aimed to reduce the number of visual tokens from $N$ to $\hat{N}$ ($\hat{N} < N$). The compressed results are represented as $\hat{E}_v = \{\hat{v}_1, \hat{v}_2, ..., \hat{v}_{\hat{N}}\}$:

$$C : E_v \Rightarrow \hat{E}_v, \tag{1}$$

where $N = |E_v|, \hat{N} = |\hat{E}_v|$ are the set size of $E_v$ and $\hat{E}_v$, respectively. $C$ is typically referred to as the patch merge layer (PML), which will be discussed in Sec3.3.

**Layer-wise Token Compression.** In previous methods, PML is typically placed downstream of the vision encoder, indicating that visual token compression is executed subsequent to the processing of the vision encoder. As illustrated in Fig.2, we further insert the PML within an intermediate $k$-th layer of the vision encoder. Specifically, assuming that the vision encoder consists of $L$ layers, from

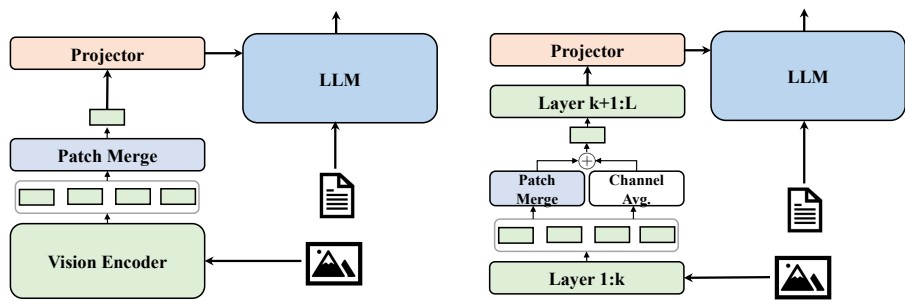

(a) Token compression after the vision encoder    (b) Layer-wise token compression with residual connection

Figure 2: Comparison between encoder-level and layer-level visual token compression. The left presents compressing tokens subsequent to the encoder, which has low efficiency, while the right demonstrates compressing tokens in the intermediate layers with PML and residual connection.

the first to the $k$-th layer, the token set $E_v$ is encoded. Immediately after the $k$-th layer, the PML converts $E_v$ to $\hat{E}_v$, with the compression ratio $r$. The compressed tokens $\hat{E}_v$ are then fed into the remaining layers from $k+1$ to $L$ for further encoding. The process is formulated as follows:

$$E_v^k = ENC_{1:k}(V)$$
$$\hat{E}_v^k = PML(E_v^k, r) \tag{2}$$
$$\hat{E}_v = ENC_{k+1:L}(\hat{E}_v^k),$$

where $ENC$ and $PML$ represent the vision encoder and patch merge layer, respectively. $E_v^k$ and $\hat{E}_v^k$ denote the original and compressed visual token sets, which satisfies $|\hat{E}_v^k| = \frac{|E_v^k|}{r^2}$.

### 3.3 PATCH MERGE LAYER (PML)

**Basic Patch Merge Layer.** Among existing visual token compression methods, we choose to utilize pixel-shuffle to implement the basic PML. Since it shows great performance in token reduction (Lu et al., 2025). The pixel-shuffle (Shi et al., 2016) operation first merges adjacent tokens with a grid. Then, a two-layer MLP maps the merged tokens back to the original embedding dimensionality:

$$\hat{E}_v = MLP(PS((E_v, r)), \tag{3}$$

where $PS$ means the pixel-shuffle operation. For instance, $E_v$ with shape $N \times C$ is first reshaped to $H \times W \times C$ and then converted to $\frac{H}{r} \times \frac{W}{r} \times r^2 C$ by pixel-shuffle, where $H \times W = N$. After reshaping back to $\frac{N}{r^2} \times r^2 C$, it is fed into the MLP and converted to $\frac{N}{r^2} \times C$, achieving the compression with ratio $r$.

**Residual Connection**. Existing visual token compression methods, such as pixel-shuffle, convolutional architectures, and TokenPacker, often suffer from the information loss problem. The problem is more pronounced when PML is placed into an intermediate layer of the vision encoder. To alleviate this limitation, motivated by Chen et al. (2024a), we propose incorporating a residual connection into the PML. Specifically, the residual pathway consists of a non-parametric shortcut implemented via a space-to-channel operation, followed by a channel averaging step to align the channel dimension. The process is formulated as:

$$\hat{E}_v = RC(E_v^k, r) = CA(PS(E_v, r)), \tag{4}$$

where $RC$ represents the residual connection and $CA$ denotes the channel averaging operation. The process is similar to the PML but introduces no parameter. Therefore, the midline in Eq.2 is updated as follows:

$$\hat{E}_v^k = PML(E_v^k, r) + RC(E_v^k, r). \tag{5}$$

The PML with residual connection achieves efficient compression while mitigating the information loss.

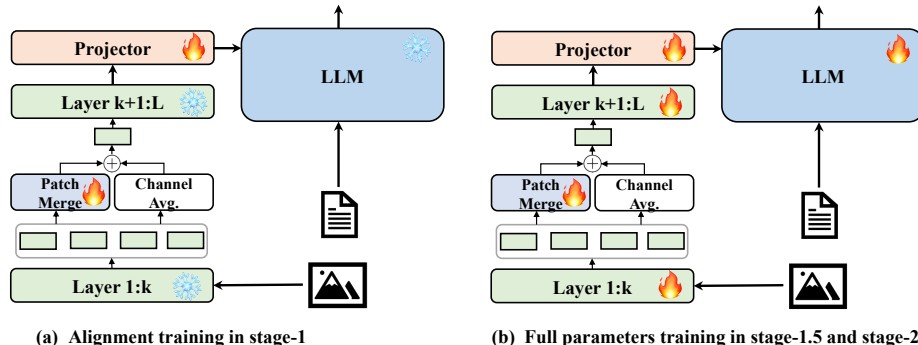

(a) **Alignment training in stage-1**    (b) **Full parameters training in stage-1.5 and stage-2**

Figure 3: Detailed overview of the training stage. The left figure represents stage-1, which only learn the projector and PML. The right part depicts the stage-1.5 and stage-2, training with all parameters.

## 3.4 TRAINING STRATEGY

We follow the training strategies proposed in LLaVA-OneVision (Li et al., 2024a), which comprises three stages and is based on the curriculum learning principle of increasing difficulty gradually.

- Stage 1: *Language-Image Alignment* focuses on aligning visual features with the textual embedding space of the language model, enabling cross-modal understanding.
- Stage 1.5: *High-Quality Knowledge Learning* aims to refine the model's generation capabilities by leveraging high-quality, curated multimodal data.
- Stage 2: *Visual Instruction Tuning* is oriented toward teaching MLLM to solve a diverse set of visual tasks with preferred responses. This stage is composed of two phases: (a) *Single-Image(SI) Training*: learning visual tasks using a single image. (b) *OneVision Training*: equipping MLLM with the abilities to process diverse inputs, such as video, single-image, and multi-image.

The first stage only learn the projector while the other two stages train MLLM with all parameters. Unlike this, due to the introduction of the inner PML, we also train the PML parameters alongside the projector during Stage 1. The remaining stages remain unchanged in our training pipeline. The training stages are depicted in Fig.3.

## 4 EXPERIMENTS

In this section, we first introduce the experiment settings. Then, we compare LaCo with other compression methods and evaluate inner-layer and external compression with various encoders. Finally, we analyze the results of placing LaCo at the different layers.

### 4.1 EXPERIMENT SETUP

**Experiments Configuration.** We use three vision encoders: AIMv2 (Fini et al., 2024), SigLIP (Zhai et al., 2023), and InternViT (Chen et al., 2024e), which are widely adopted in MLLMs. The projector architecture follows the design employed in LLaVA-OneVision. We utilize Qwen2.5-0.5B as the LLM with consideration of the compute budget. All models are trained for 1 epoch with a batch size of 512. In Stage-1, the learning rate is set to be 1e-3. In both Stage 1.5 and Stage 2, the vision encoder is fine-tuned with a learning rate of 2e-6, while the remaining modules are optimized using a learning rate of 1e-5. We adopt the cosine learning rate schedule with the minimum value of 1e-7.

**Baselines and Models.** For baseline comparisons, we implement LaCo and three representative approaches, including Pixel-Shuffle (Chen et al., 2024d), TokenPacker (Li et al., 2024e) and the LDPv2 (Chu et al., 2024). All methods are employed at the 1/4 layer of the AIMv2. We also compare inner-layer and external compression across three widely used vision encoders involving AIMv2 (Fini et al., 2024), SigLIP (Zhai et al., 2023), and InternViT (Chen et al., 2024e), with

Table 1: LaCo outperforms Pixel-Shuffle (Chen et al., 2024d), LDPv2 (Chu et al., 2024), and To-kenPacker (Li et al., 2024e) on single-image benchmarks. LaCo achieves comparable results to LLaVA-OV-0.5B (Li et al., 2024a) with significantly improved efficiency. All compression methods use AIMv2 as the vision encoder and are placed at the 1/4 layer.

| Method | PT | IT | VET | TPS | AI2D | ChartQA | DocVQA | InfoVQA | MMBench | MMMU | MMStar | S-Bench | S-QA | RW-QA | Avg. |
|---|---|---|---|---|---|---|---|---|---|---|---|---|---|---|---|
| *Single-Image Training Setting* | | | | | | | | | | | | | | | |
| LLaVA-OV-0.5B | - | - | 138.4 | 18.5 | 54.2 | 61.0 | 75.0/71.2 | 44.8/41.3 | 43.8 | 31.2 | 36.3 | 63.4 | 67.8 | 53.7 | 52.8 |
| Pixel-Shuffle | **10.8** | **18.4** | 29.0 | 27.4 | 56.5 | 13.7 | 13.1/13.0 | 19.4/20.0 | 36.9 | 32.7 | 30.6 | 50.9 | 66.5 | 43.5 | 36.4 |
| LDPv2 | 10.9 | 18.6 | 30.1 | 27.4 | 56.3 | 13.9 | 12.4/13.0 | 19.3/20.0 | 35.2 | 33.4 | 30.3 | 49.3 | 64.7 | 45.0 | 36.0 |
| TokenPacker | 11.5 | 20.0 | 33.6 | 26.8 | 56.3 | 13.6 | 12.6/13.0 | 19.2/20.0 | 35.0 | 32.0 | 30.7 | 49.9 | 66.5 | 44.7 | 36.1 |
| LaCo | **10.8** | 18.6 | 29.3 | **27.4** | **59.3** | **55.9** | **63.8/65.0** | **33.6/34.0** | **55.2** | **35.7** | **37.7** | **64.7** | **69.2** | **55.2** | **53.1** |
| *One-Vision Training Setting* | | | | | | | | | | | | | | | |
| LLaVA-OV-0.5B | - | - | 137.8 | 18.3 | 57.1 | **61.4** | **73.7/70.0** | **46.3/41.8** | 52.1 | 31.4 | 37.5 | **65.5** | 67.2 | 55.6 | **54.4** |
| Pixel-Shuffle | **10.8** | 14.1 | **29.4** | 27.5 | 57.6 | 14.0 | 12.0/12.0 | 20.4/20.0 | 38.8 | 34.1 | 30.7 | 52.2 | 59.3 | 44.4 | 36.3 |
| LDPv2 | 10.9 | **13.7** | 29.7 | 27.6 | 57.9 | 14.4 | 12.0/12.0 | 20.5/20.0 | 36.7 | 33.6 | 28.6 | 51.0 | 62.5 | 45.5 | 36.2 |
| TokenPacker | 11.5 | 14.7 | 33.6 | 26.9 | 58.5 | 14.5 | 12.2/12.0 | 20.0/20.0 | 38.1 | 33.7 | 29.6 | 51.9 | 60.6 | 46.5 | 36.6 |
| LaCo | **10.8** | 13.9 | 29.5 | **27.6** | **61.8** | 59.2 | 62.4/64.0 | 34.7/34.0 | **58.7** | **35.2** | **39.5** | 62.0 | 66.7 | 55.4 | 53.6 |

LaCo applied for token compression. Furthermore, to explore the effects of compression at different layers, we place LaCo at the 1/12, 1/6, 1/4, and 1/2 layers of the vision encoder, exemplified by the AIMv2.

**Training Datasets.** We follow the training datasets as LLaVA-OneVision (Li et al., 2024a). In the pretrained stage-1, the LCS-558K (Liu et al., 2023b) is used to tune the projector and the PML. In the stage-1.5, we utilize the 4M high-quality knowledge data, including re-captioned detailed description data, document/OCR data, and Chinese language data. The visual instruction tuning data for stage-2 also comes from LLaVA-OneVision, comprising 3.2M single-image instruction data in SI phase and 1.6M mixed data in one-vision phase. For unpublished data, we adopt similar data under the training guidance of LLaVA-OneVision.

## 4.2 MAIN RESULTS

To validate the effectiveness of our proposed LaCo, we compare it with three representative visual token compression approaches, including Pixel-Shuffle (Chen et al., 2024d), LDPv2 (Chu et al., 2024), and TokenPacker (Li et al., 2024e). All methods are applied at the 1/4 layer of the AIMv2 vision encoder, with Qwen2.5-0.5B used as the language model.

**Evaluation on Single-Image Benchmarks.** As demonstrated in Tab.1, LaCo consistently outper-forms the comparison methods across all evaluated scenarios. Specifically, compared to LaCo, Pixel-Shuffle, LDPv2, and TokenPacker exhibit performance dropping by over 30% in terms of average metrics in both single-training and one-vision training. Although they demonstrate compet-itive performance when applied externally, their effectiveness diminishes significantly when used to compress tokens in the intermediate layers of vision encoders, highlighting the superiority of our layer-wise design. LaCo further improves upon Pixel-Shuffle by introducing residual connections, which play a critical role in preserving informative visual features during compression. Notably, on challenging benchmarks such as DocVQA (Mathew et al., 2020), InfoVQA (Mathew et al., 2021), and MMBench (Liu et al., 2023d), where accurate task completion relies on rich contextual informa-tion, Pixel-Shuffle, LDPv2, and TokenPacker suffer from performance degradation due to substan-tial information loss. This further indicates the effectiveness of LaCo in preserving critical visual content during token compression. Moreover, we present extensive case studies in Appendix A to demonstrate the superiority of LaCo in handling diverse complex tasks.

We also report the efficiency of these methods in terms of **Pre-training Time (PT)**, **Instruction Tuning Time (IT)**, **Vision Encoding Time (VET)**, and **Tokens Per Second (TPS)**, where PT and IT evaluate training efficiency, VET and TPS quantify inference efficiency. TokenPacker exhibits the lowest overall efficiency, while LDPv2, Pixel-shuffle, and LaCo demonstrate comparable per-formance in terms of efficiency. However, LaCo achieves significantly better results in effectiveness while maintaining high efficiency.

Table 2: LaCo outperforms Pixel-Shuffle (Chen et al., 2024d), LDPv2 (Chu et al., 2024), Token-Packer (Li et al., 2024e), and LLaVA-OV-0.5B (Li et al., 2024a) on multi-image benchmarks.

| Method | IEI | MI-VQA | NL-VR2 | Puzzle | Q-Bench | Spot-Diff | TR-VQA | VST | 3D-Chat | 3D-TD | ScanQA | ALFRED | nuScenes | BLINK | Mantis | MathVerse | MuirBench | SciVerse | Avg. |
|---|---|---|---|---|---|---|---|---|---|---|---|---|---|---|---|---|---|---|---|
| | | | in-domain multi-image | | | | | | | in-domain multi-view | | | | | | out-domain | | | |
| LLaVA-OV-0.5B | 17.1 | 48.7 | 63.4 | 35.4 | 48.8 | 36.4 | **65.0** | 29.8 | 60.0 | 48.0 | 26.4 | **62.2** | 70.5 | **52.1** | 39.6 | **60.0** | 25.5 | 29.1 | 45.4 |
| Pixel-Shuffle | 27.8 | 72.3 | 63.7 | 35.2 | 47.4 | **38.9** | 58.0 | 29.3 | 60.5 | 48.7 | 26.1 | 60.2 | 57.5 | 38.8 | 38.6 | 26.0 | 27.4 | 26.7 | 43.5 |
| LDPv2 | 27.3 | 76.2 | 61.9 | **46.1** | **49.0** | 37.2 | 60.1 | 28.6 | 60.5 | 48.5 | 25.2 | 60.3 | 64.2 | 38.2 | 36.1 | 24.4 | 26.2 | 22.0 | 44.0 |
| TokenPacker | 27.6 | 73.3 | 61.7 | 44.0 | 47.8 | 36.8 | 56.9 | 28.7 | 60.4 | 48.5 | 26.9 | 59.7 | 58.2 | 38.3 | 34.3 | 25.5 | 26.4 | 25.1 | 43.3 |
| LaCo | **30.0** | **85.0** | **73.2** | 43.9 | 48.5 | 38.8 | 60.5 | **30.6** | 60.5 | **48.9** | **30.3** | 59.7 | **72.5** | 41.3 | **44.4** | 34.3 | **31.9** | **30.4** | **48.0** |

**Evaluation on Multi-Image Benchmarks.** For multi-image benchmarks, the models evaluated are identical to those used in the single-image setting. Therefore, we omit the efficiency metrics in Tab.2 for brevity. As shown in the table, LaCo achieves superior performance compared to methods Pixel-Shuffle, LDPv2, and TokenPacker, surpassing them by 10.3%, 9.1%, and 10.9% respectively in average benchmark metrics. This fully demonstrates the effectiveness of LaCo in handling complex tasks such as multi-image reasoning, identifying differences, and understanding 3D environments.

Table 3: LaCo outperforms Pixel-Shuffle (Chen et al., 2024d), LDPv2 (Chu et al., 2024), Token-Packer (Li et al., 2024e), and LLaVA-OV-0.5B (Li et al., 2024a) on video benchmarks.

| Method | EgoSchema | NextQA | PercepTest | S-bench | VideoMME | Avg. |
|---|---|---|---|---|---|---|
| | test | mc | val | video | wo/w-subs | |
| LLaVA-OV-0.5B | 26.8 | 57.2 | **49.2** | 44.2 | **44.0**/43.5 | 44.2 |
| Pixel-Shuffle | 21.4 | 44.8 | 42.7 | 33.0 | 36.1/42.4 | 36.2 |
| LDPv2 | 21.6 | 42.6 | 41.6 | 31.2 | 34.9/41.3 | 35.0 |
| TokenPacker | 22.4 | 45.2 | 41.2 | 31.6 | 36.1/41.8 | 35.9 |
| LaCo | **29.6** | **57.8** | 47.4 | **44.5** | 41.0/**46.0** | **44.6** |

**Evaluation on Video Benchmarks.** Tab.3 demonstrates the results on video benchmarks. LaCo outperforms Pixel-Shuffle, LDPv2, and TokenPacker by 23.2%, 27.4%, and 24.2%, respectively, in comparative evaluations. The results highlight the superiority of LaCo in handling temporal-spatial information.

As shown in Tables 1-3, LaCo exhibits significant inference efficiency advantages over LLaVA-OV while maintaining competitive performance. With only 25% of the token consumption required by LLaVA-OV, LaCo achieves comparable results on single-image benchmarks and outperforms LLaVA-OV in both multi-image and video benchmarks. More critically, LaCo improves VET by 78.6% and TPS by 50.8%. Appendix C compares LaCo with state-of-the-art multimodal models on larger architectures.

## 4.3 GENERALIZATION TO DIFFERENT ENCODERS

To evaluate the generalization capability and efficiency improvements of our approach, we further conduct experiments using three vision encoders involving AIMv2, SigLIP, and InternViT (referred to as InViT). We compare token compression performed within an intermediate layer of the vision encoder to compression applied externally. Specifically, we apply our proposed LaCo both at the 1/4 layer of the encoder and subsequent to the encoder. Qwen2.5-0.5B is used as the LLM.

As shown in Tab.4, AIMv2-LaCo@1/4 reduces PT, IT, and VET by 7.7%, 26.5%, and 48.8% respectively, while increasing TPS by 15.0% compared to AIMv2-LaCo@1 (LaCo applied subsequent to the encoder). Similarly, when applied to SigLIP and InternVIT, the training and inference efficiency improvements reach 29.3%-26.6% and 22.8%–30.5%, respectively. In terms of effectiveness, the results show mixed outcomes. For SigLIP, inner-layer compression (LaCo@1/4) outperforms external compression by 2.7 in average accuracy. AIMv2 and InternViT both exhibit slight declines, decreasing by 5.0% and 4.9%, respectively. Specifically, performance drops much only on DocVQA, InfoVQA, and ChartQA benchmarks while even gets improved on ScienceQA and RealWordQA. We attribute this phenomenon to the fact that tasks on DocVQA, InfoVQA, and ChartQA require rich and fine-grained visual information to achieve good results. Inner-layer compression inevitably

Table 4: Performance comparison between LaCo@1/4 and LaCo@1 in AIMv2 (Fini et al., 2024), SigLIP (Zhai et al., 2023), and InternViT (Chen et al., 2024e) on single-image benchmarks. Layer-wise token compression achieves comparable results with significantly improved efficiency. LaCo@x means placing LaCo at the x-th layer of the vision encoder.

| Model | PT | IT | VET | TPS | AI2D | ChartQA | DocVQA | InfoVQA | MMBench | MMMU | MMStar | S-Bench | S-QA | RW-QA | Avg. |
|---|---|---|---|---|---|---|---|---|---|---|---|---|---|---|---|
| *Single-Image Training Setting* | | | | | | | | | | | | | | | |
| SigLIP-LaCo@1 | 14.5 | 29.6 | 146.5 | 18.0 | 56.5 | 13.0 | 12.2/12.0 | 19.9/20.0 | 27.1 | 32.8 | 27.2 | 44.0 | 62.5 | 46.7 | 34.2 |
| SigLIP-LaCo@1/4 | 11.9 | 22.2 | 57.3 | 24.0 | 56.8 | 13.3 | 12.3/13.0 | 19.6/20.0 | 40.1 | 33.1 | 33.2 | 55.4 | 65.1 | 45.2 | 37.5 |
| InViT-LaCo@1 | 14.0 | 28.6 | 107.6 | 19.0 | 60.8 | 67.5 | 76.6/76.0 | 41.7/41.1 | 56.1 | 36.2 | 38.6 | 65.2 | 70.3 | 54.2 | **56.7** |
| InViT-LaCo@1/4 | 12.3 | 22.3 | 43.4 | 25.5 | 59.2 | 59.7 | 68.8/70.0 | 34.6/35.0 | 52.9 | 34.8 | **38.9** | 62.6 | 69.2 | 51.1 | 53.3 |
| AIMv2-LaCo@1 | 11.7 | 25.8 | 58.0 | 24.0 | **60.9** | 63.2 | 73.5/74.0 | 39.6/39.0 | 55.6 | 35.1 | 38.6 | **66.4** | **70.7** | **56.1** | 56.0 |
| AIMv2-LaCo@1/4 | **10.8** | **19.3** | **29.3** | **27.4** | 59.3 | 55.9 | 63.8/65.0 | 33.6/34.0 | 55.2 | 35.7 | 37.7 | 64.7 | 69.2 | 55.2 | 53.1 |
| *One-Vision Training Setting* | | | | | | | | | | | | | | | |
| SigLIP-LaCo@1 | 14.5 | 26.5 | 146.3 | 18.2 | 57.4 | 13.8 | 11.8/12.0 | 20.1/20.0 | 34.6 | 34.4 | 30.8 | 47.1 | 52.5 | 45.0 | 34.8 |
| SigLIP-LaCo@1/4 | 11.9 | 15.8 | 57.4 | 24.8 | 57.9 | 15.9 | 11.7/12.0 | 20.1/20.0 | 47.9 | 34.4 | 33.8 | 57.8 | 60.4 | 46.0 | 38.6 |
| InViT-LaCo@1 | 14.0 | 23.2 | 107.7 | 19.0 | 62.5 | 69.5 | 75.3/76.0 | 41.2/41.0 | 57.9 | 34.9 | 40.7 | 66.9 | 64.5 | 54.5 | **56.9** |
| InViT-LaCo@1/4 | 12.3 | 16.2 | 43.7 | 24.8 | 61.0 | 61.7 | 66.9/68.0 | 36.1/36.0 | 54.9 | **36.2** | 38.9 | 65.3 | **67.6** | 52.2 | 54.1 |
| AIMv2-LaCo@1 | 11.7 | 18.9 | 57.6 | 24.0 | 62.2 | 65.6 | 71.8/73.0 | 40.1/41.0 | **60.1** | **36.2** | **40.9** | **67.9** | 64.3 | 54.2 | 56.4 |
| AIMv2-LaCo@1/4 | **10.8** | **13.9** | **29.5** | **27.6** | 61.8 | 59.2 | 62.4/64.0 | 34.7/34.0 | 58.7 | 35.2 | 39.5 | 62.0 | 66.7 | **55.4** | 53.6 |

Table 5: Performance comparison between LaCo@1/4 and LaCo@1 in AIMv2 (Fini et al., 2024), SigLIP (Zhai et al., 2023), and InternViT (Chen et al., 2024e) on multi-image benchmarks.

| Model | IEI | MI-VQA | NL-VR2 | Puzzle | Q-Bench | Spot-Diff | TR-VQA | VST | 3D-Chat | 3D-TD | ScanQA | ALFRED | nuScenes | BLINK | Mantis | MathVerse | MuirBench | SciVerse | Avg. |
|---|---|---|---|---|---|---|---|---|---|---|---|---|---|---|---|---|---|---|---|
| | | | in-domain multi-image | | | | | | | in-domain multi-view | | | | | | out-domain | | | |
| SigLIP-LaCo@1 | 26.9 | 67.5 | 58.1 | 35.4 | 48.2 | 34.8 | 57.0 | 29.2 | 60.5 | 48.4 | 19.8 | 58.0 | 55.8 | 38.7 | 42.2 | 25.0 | 24.9 | 25.6 | 42.0 |
| SigLIP-LaCo@1/4 | 28.4 | 78.0 | 67.4 | 28.9 | 48.3 | 37.3 | 58.2 | 29.6 | 60.5 | 48.7 | 25.5 | 59.6 | 62.2 | 38.5 | 39.0 | 30.8 | 31.3 | 26.0 | 44.3 |
| InViT-LaCo@1 | 29.4 | 81.0 | 73.8 | 37.6 | 49.5 | **40.1** | 64.6 | 30.3 | **60.8** | **49.1** | 30.1 | 57.5 | 72.8 | 40.5 | 43.5 | 36.3 | 34.8 | 40.0 | 48.4 |
| InViT-LaCo@1/4 | 29.3 | 80.2 | 72.3 | 36.9 | 47.3 | 37.9 | 61.6 | 30.2 | **60.8** | 48.8 | 30.0 | 56.3 | **75.3** | 39.6 | 43.5 | **44.8** | 33.2 | **49.6** | 48.8 |
| AIMv2-LaCo@1 | **30.4** | 82.8 | **75.7** | 42.4 | **49.9** | 37.3 | **66.1** | 30.2 | 60.6 | **49.1** | 31.3 | 59.5 | 73.0 | 40.5 | **44.9** | 42.5 | **35.9** | 41.8 | **49.7** |
| AIMv2-LaCo@1/4 | 30.0 | **85.0** | 73.2 | **43.9** | 48.5 | 38.8 | 60.5 | **30.6** | 60.5 | 48.9 | 30.3 | **59.7** | 72.5 | **41.3** | 44.4 | 34.3 | 31.9 | 30.4 | 48.0 |

reduces more information than external compression, thus performing worse. However, it is still useful to apply inner-layer token compression when the compute budget is limited.

Table 6: Performance comparison between LaCo@1/4 and LaCo@1 in AIMv2 (Fini et al., 2024), SigLIP (Zhai et al., 2023), and InternViT (Chen et al., 2024e) on video benchmarks.

| Model | EgoSchema | NextQA | PercepTest | S-bench | VideoMME | Avg. |
|---|---|---|---|---|---|---|
| | test | mc | val | video | wo/w-subs | |
| SigLIP-LaCo@1 | 20.6 | 40.4 | 41.6 | 31.4 | 36.1/41.0 | 34.5 |
| SigLIP-LaCo@1/4 | 23.3 | 49.6 | 43.2 | 34.6 | 37.3/44.3 | 37.9 |
| InViT-LaCo@1 | 29.0 | 61.4 | 46.8 | 45.8 | **43.6**/47.8 | 45.7 |
| InViT-LaCo@1/4 | 27.8 | 59.9 | 46.9 | 40.6 | 42.2/46.6 | 43.9 |
| AIMv2-LaCo@1 | **30.0** | **61.6** | **47.9** | **47.4** | 43.1/**48.1** | **46.5** |
| AIMv2-LaCo@1/4 | 29.6 | 57.8 | 47.4 | 44.5 | 41.0/46.0 | 44.6 |

For multi-image benchmarks, Tab.5 shows the detailed results. As seen from the table, the performance gap between inner-layer and external compression narrows significantly under the multi-image setup, which can be verified on both AIMv2, SigLIP, and InternViT encoders. Among the three encoders, AIMv2 gets the best performance while SigLIP ranks relatively lower.

Tab.6 demonstrates the results on video benchmarks. While inner-layer compression still has room for improvement compared to external token compression, it delivers competitive performance while significantly improving training efficiency. This suggests that LaCo has strong potential for deployment in resource-constrained scenarios involving long visual sequences such as videos.

## 4.4 IMPACT OF LACO PLACEMENT DEPTH

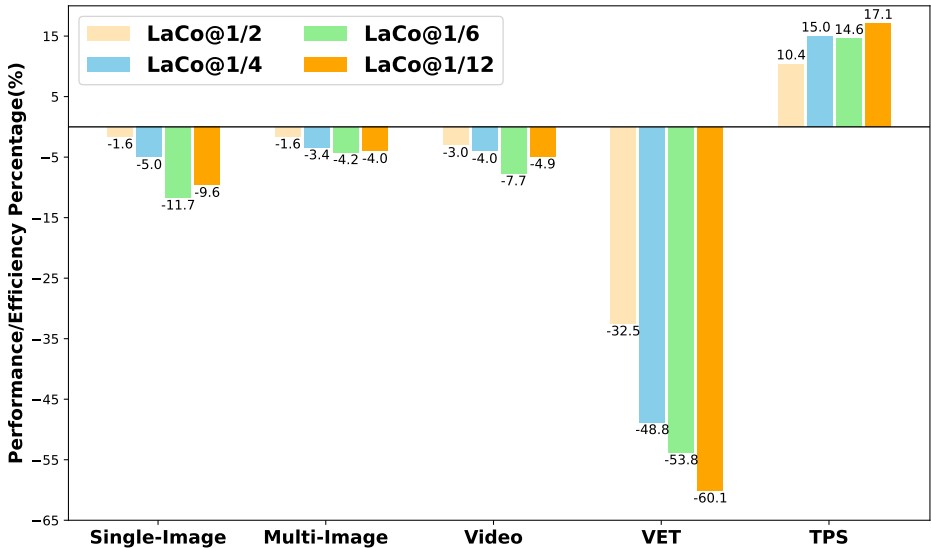

Figure 4: Performance and efficiency percentage variations of internal versus external token compression within AIMv2. LaCo@x means placing LaCo at the x-th layer of AIMv2.

To further investigate the trade-off between performance and efficiency when placing LaCo at different depths of the vision encoder, we conduct experiments using AIMv2 as the vision encoder and Qwen2.5-0.5B as the language model. Specifically, LaCo is inserted at four distinct layers: 1/12, 1/6, 1/4, and 1/2 of the total encoder depth.

As shown in Fig.4, the first three bar groups summarize the relative performance drop across different compression layers. The last two groups illustrate the corresponding improvement in inference efficiency. The results show that compressing vision tokens within intermediate layers improves TPS by over 10% and reduces VET up to 60%, despite some performance degradation. Moreover, as the compression layer decreases, performance drops more rapidly, whereas inference efficiency gains become increasingly marginal. It's a good idea to choose to compress tokens in the 1/2 layer of the vision encoder, striking a balance between performance and efficiency. It is worth noting that compressing tokens at the 1/12 layer achieves higher performance compared to compression at the 1/6 layer. This indicates that early compression preserves more useful visual information when compressing at shallower layers ($\leq 1/6$) of the vision encoder. In addition, we give the detailed evaluation results of layer-wise and external token compression in Appendix B.

## CONCLUSIONS

We demonstrate that compressing visual tokens within the intermediate layers of the vision encoder holds significant potential for improving the efficiency of MLLMs. To this end, we propose LaCo, a novel token compression method that leverages pixel-shuffle operations and residual connections to merge visual tokens with minimal information loss. Experimental results validate LaCo's effectiveness, showing over a 20% improvement in training efficiency and a 15% increase in inference throughput while maintaining strong performance—surpassing existing methods in terms of information preservation. Nonetheless, several directions remain open for future exploration. For instance, investigating adaptive compression ratios($r$) across different layers or tasks could further enhance flexibility and performance. We plan to explore these avenues in our future work.

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

## A  EVALUATION BENCHMARKS.

We evaluate our model on three different benchmark groups:

- **Single-Image Benchmarks:** including (a) *Chart, Diagram, and Document Understanding:* AI2D (Kembhavi et al., 2016), ChartQA (Masry et al., 2022), DocVQA (Mathew et al., 2020), and InfoVQA (Mathew et al., 2021); (b) *Perception and Multi-discipline Reasoning:* MMBench (Liu et al., 2023d), MMMU (Yue et al., 2024), MMStar (Chen et al., 2024c), SeedBench(image) (Li et al., 2023a), and ScienceQA(S-QA) (Saikh et al., 2022); (c) *Real-world Understanding and Visual Chat:* RealWorldQA(RW-QA) (xAI, 2024). It consists of 10 benchmarks.

- **Multi-Image Benchmarks:** including (a) *in-domain multi-image:* Spot the Difference (Jhamtani & Berg-Kirkpatrick, 2018), Image Edit Instruction(IEI) (Li et al., 2024c), Visual Storytelling(VST) (Ting-Hao et al., 2016), Text-rich VQA(TR-VQR) (Liu et al., 2023c), Multi-image VQA(MI-VQA) (Raj et al., 2021), Raven Puzzle (Chia et al., 2024), Q-Bench (Wu et al., 2023), and NLVR2 (Suhr & Artzi, 2019); (b) *in-domain multi-view:*

3D Dialogue (3D-Chat) and Task Decomposition (3D-TD) from 3D-LLM (Hong et al., 2023), ScanQA (Azuma et al., 2021), ALFRED (Shridhar et al., 2019), and nuScenes (Caesar et al., 2019); (c) *out-domain tasks:* BLINK (Fu et al., 2024b), MMMU(multi-image) (Yue et al., 2024), MuirBench (Wang et al., 2024a), and MathVerse (Zhang et al., 2024a). It comprises 18 benchmarks.

- **Video Benchmarks:** containing 5 benchmarks: EgoSchema (Mangalam et al., 2023), NeXTQA (Xiao et al., 2021), PerceptionTest (Puatruaucean et al., 2023), Seed-Bench(video) (Li et al., 2023a), and VideoMME (Fu et al., 2024a).

# B  VISUAL RESULTS

To further validate the practical effectiveness of our approach in visual comprehension, we evaluate it on diverse real-world tasks requiring complex reasoning, as demonstrated in Fig.5. Equipped with pixel-shuffle and residual connections, LaCo demonstrates superior performance across challenging scenarios, consistently outperforming existing methods like Pixel-Shuffle, LDPv2, and Token-Packer. This aligns with findings that information-aware designs enhance practical applicability in multimodal tasks, particularly in tasks demanding rich contextual understanding.

# C  DETAILED EVALUATION OF LAYER-WISE COMPRESSION

We detail the evaluation of the experimental results applying our proposed LaCo at the different layers of AIMv2. The results are demonstrated on Tab.7, Tab.8, and Tab.9, which represent the model performance on single-image, multi-image, and video benchmarks, respectively.

Table 7: Performance and efficiency comparison between LaCo@1/12, LaCo@1/6, LaCo@1/4, LaCo@1/2 and LaCo@1 on single-image benchmarks.

| Method | PT | IT | VET | TPS | AI2D | ChartQA | DocVQA | InfoVQA | MMBench | MMMU | MMStar | S-Bench | S-QA | RW-QA | Avg. |
|---|---|---|---|---|---|---|---|---|---|---|---|---|---|---|---|
| *Single-Image Training Setting* | | | | | | | | | | | | | | | |
| LaCo@1/12 | 10.9 | **18.0** | **23.5** | **28.2** | 59.1 | 50.6 | 53.9/55.0 | 28.2/29.0 | 50.9 | 34.6 | 39.2 | 62.5 | 69.5 | 53.3 | 50.3 |
| LaCo@1/6 | 10.9 | 18.5 | 26.2 | 27.7 | 59.5 | 46.9 | 51.7/54.0 | 25.7/29.0 | 45.3 | 36.0 | 36.2 | 61.8 | 68.4 | 50.5 | 48.5 |
| LaCo@1/4 | **10.8** | 19.3 | 29.3 | 27.4 | 59.3 | 55.9 | 63.8/65.0 | 33.6/34.0 | 55.2 | 35.7 | 37.7 | 64.7 | 69.2 | 55.2 | 53.1 |
| LaCo@1/2 | 11.0 | 20.0 | 39.0 | 26.3 | 59.3 | 60.8 | 68.1/70.0 | 38.5/**39.0** | **56.9** | **36.4** | 37.5 | 65.5 | 70.6 | 54.2 | 54.9 |
| LaCo@1 | 11.7 | 25.8 | 58.0 | 24.0 | **60.9** | **63.2** | **73.5/74.0** | **39.6/39.0** | 55.6 | 35.1 | **38.6** | 66.4 | **70.7** | **56.1** | **56.0** |
| *One-Vision Training Setting* | | | | | | | | | | | | | | | |
| LaCo@1/12 | 10.9 | **13.4** | **23.0** | **28.1** | 61.0 | 53.4 | 51.6/53.0 | 29.0/29.0 | 54.9 | 36.0 | 37.5 | 65.7 | 65.9 | 53.9 | 51.0 |
| LaCo@1/6 | 10.9 | 13.7 | 26.6 | 27.5 | 60.6 | 51.7 | 51.8/54.0 | 28.1/29.0 | 54.3 | 35.1 | 37.1 | 64.4 | 65.2 | 48.5 | 49.8 |
| LaCo@1/4 | **10.8** | 13.9 | 29.5 | 27.6 | 61.8 | 59.2 | 62.4/64.0 | 34.7/34.0 | 58.7 | 35.2 | 39.5 | 62.0 | 66.7 | **55.4** | 53.6 |
| LaCo@1/2 | 11.0 | 14.5 | 38.9 | 26.5 | 61.8 | 62.0 | 67.0/68.0 | 38.9/39.0 | **60.1** | 35.8 | 39.1 | 67.6 | **66.9** | 55.0 | 55.5 |
| LaCo@1 | 11.7 | 18.9 | 57.6 | 24.0 | **62.2** | **65.6** | **71.8/73.0** | **40.1/41.0** | **60.1** | **36.2** | **40.9** | 67.9 | 64.3 | 54.2 | **56.4** |

Table 8: Performance comparison between LaCo@1/12, LaCo@1/6, LaCo@1/4, LaCo@1/2 and LaCo@1 on multi-image benchmarks.

| Method | IEI | MI-VQA | NL-VR2 | Puzzle | Q-Bench | Spot-Diff | TR-VQA | VST | 3D-Chat | 3D-TD | ScanQA | ALFRED | nuScenes | BLINK | Mantis | MathVerse | MuirBench | SciVerse | Avg. |
|---|---|---|---|---|---|---|---|---|---|---|---|---|---|---|---|---|---|---|---|
| | | in-domain multi-image | | | | | | | | in-domain multi-view | | | | | out-domain | | | | |
| LaCo@1/12 | 30.0 | 79.7 | 73.4 | **44.0** | 49.0 | 38.0 | 60.3 | 30.6 | **60.8** | 49.0 | 30.4 | 56.5 | 70.0 | 39.7 | **45.8** | 34.6 | 31.0 | 35.1 | 47.7 |
| LaCo@1/6 | 29.4 | **87.3** | 72.7 | 35.9 | 48.4 | **39.3** | 61.5 | **30.9** | 60.5 | 48.9 | 30.0 | **60.1** | 68.2 | **41.3** | 43.9 | 35.5 | 31.4 | 31.8 | 47.6 |
| LaCo@1/4 | 30.0 | 85.0 | 73.2 | 43.9 | 48.5 | 38.8 | 60.5 | 30.6 | 60.5 | 48.9 | 30.3 | 59.7 | 72.5 | **41.3** | 44.4 | 34.3 | 31.9 | 30.4 | 48.0 |
| LaCo@1/2 | 29.9 | 84.8 | **76.1** | 43.6 | 48.1 | 36.4 | 65.0 | 30.1 | 60.7 | 48.9 | **31.3** | 59.2 | **75.0** | 41.2 | **45.8** | 32.6 | 31.7 | 40.0 | 48.9 |
| LaCo@1 | **30.4** | 82.8 | 75.7 | 42.4 | **49.9** | 37.3 | **66.1** | 30.2 | 60.6 | **49.1** | **31.3** | 59.5 | 73.0 | 40.5 | 44.9 | **42.5** | **35.9** | **41.8** | **49.7** |

Specifically, Tab.7 shows the performance of applying LaCo at different layers of AIMv2 on single-image benchmarks. From the table, we observe that compressing tokens within the encoder leads to a training and inference efficiency improvement exceeding 20% and 10%, respectively. Furthermore, as the compression layer is placed closer to the input (i.e., at shallower layers), the acceleration

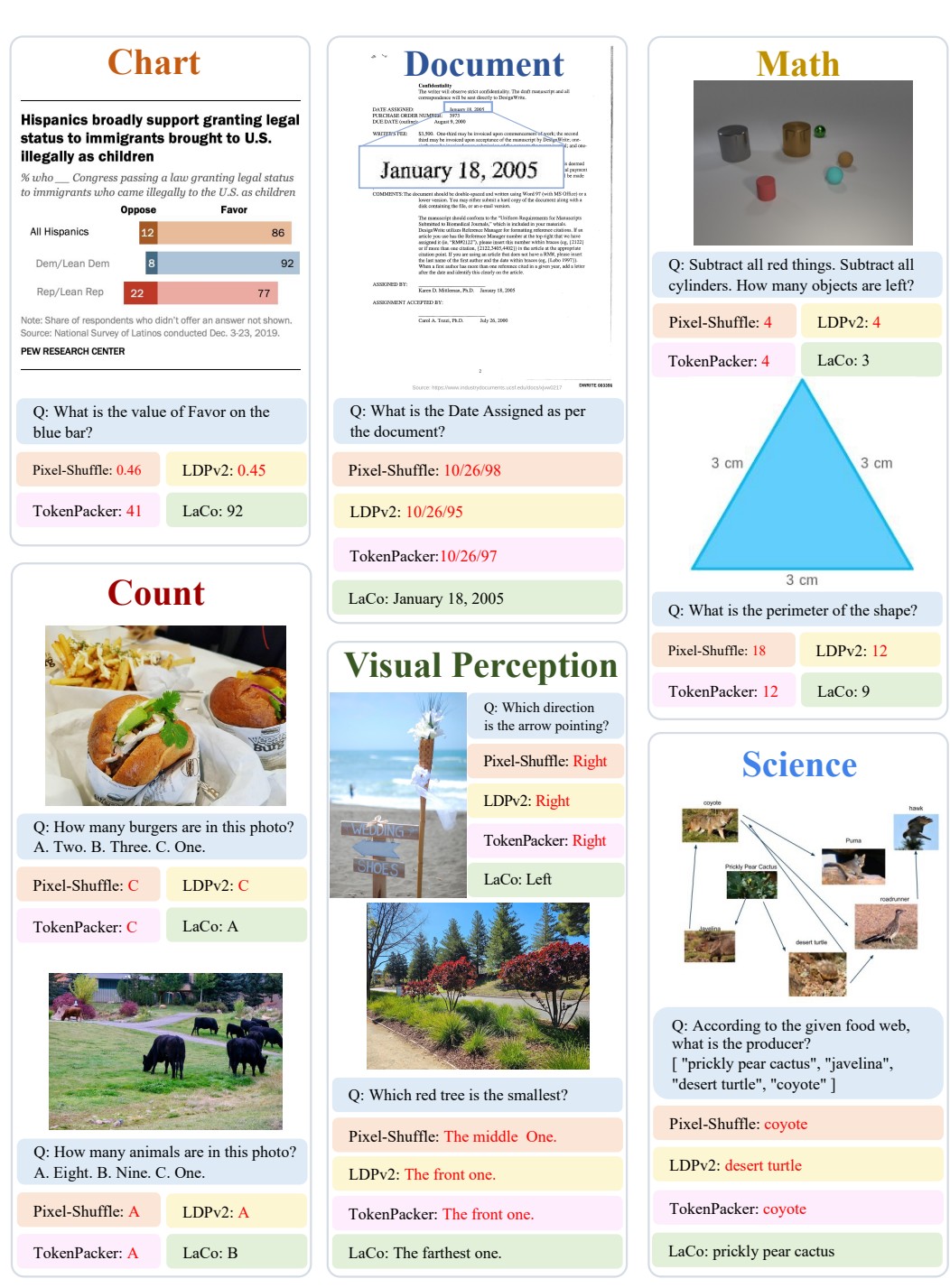

Figure 5: Visual comparison between Pixel-Shuffle (Chen et al., 2024d), LDPv2 (Chu et al., 2024), TokenPacker (Li et al., 2024e) and LaCo. Red fonts denote the wrong answers.

effect becomes more pronounced. Compared to external token compression, when performing token compression within the encoder, the model exhibits a more significant performance drop on three benchmarks: ChartQA, DocVQA, and InfoVQA. However, its performance remains comparable or even surpasses that of external compression on other benchmarks, particularly showing improved results on benchmarks MME and RealWorldQA. This is because tasks ChartQA, DocVQA, and InfoVQA require detailed information for reasoning, and internal compression inevitably leads to some loss of information, resulting in performance degradation.

Table 9: Performance comparison between LaCo@1/12, LaCo@1/6, LaCo@1/4, LaCo@1/2 and LaCo@1 on video benchmarks.

| Method | EgoSchema | NextQA | PercepTest | S-bench | VideoMME | Avg. |
|---|---|---|---|---|---|---|
| | test | mc | val | video | wo/w-subs | |
| LaCo@1/12 | 29.5 | 59.5 | 46.7 | 41.7 | 40.6/46.1 | 44.2 |
| LaCo@1/6 | 26.2 | 57.3 | 46.6 | 42.3 | 39.2/45.3 | 42.9 |
| LaCo@1/4 | 29.6 | 57.8 | 47.4 | 44.5 | 41.0/46.0 | 44.6 |
| LaCo@1/2 | 29.1 | 59.8 | 47.6 | 44.3 | 42.7/46.7 | 45.1 |
| LaCo@1 | **30.0** | **61.6** | **47.9** | **47.4** | **43.1/48.1** | **46.5** |

As shown in Tab.8, while layer-wise token compression still has room for improvement compared to external compression, it achieves competitive performance on most multi-image benchmarks. Internal compression exhibits a significant performance drop only on MathVerse, MuirBench, and SciVerse, which are relatively more challenging and require rich information for reasoning. Tab.9 demonstrates the performance on video benchmarks. Compared to external compression, layer-wise compression achieves comparable results.

# D EVALUATION ON LARGER MLLMs

Table 10: Performance comparison between LaCo-7B and other MLLMs on single-image benchmarks.

| Method | VET | TPS | AI2D | ChartQA | DocVQA | InfoVQA | MMBench | MMMU | MMStar | S-Bench | S-QA | RW-QA | Avg. |
|---|---|---|---|---|---|---|---|---|---|---|---|---|---|
| GPT-4V (OpenAI, 2023) | - | - | 78.2 | 78.5 | -/88.4 | - | 75.0 | 56.8 | 57.1 | 49.9 | 75.7 | 61.4 | - |
| GPT-4o (OpenAI, 2024) | - | - | 94.2 | 85.7 | -/92.8 | - | - | 69.1 | - | 76.2 | - | 58.6 | - |
| InternVL-2-8B (Chen et al., 2024e) | - | - | 83.8 | 83.3 | -/91.6 | -/74.8 | 81.7 | 49.3 | 59.4 | 76.0 | 97.0 | 64.4 | 76.1 |
| LLaVA-OV-7B (Li et al., 2024b) | 138.9 | 15.7 | 81.4 | 80.0 | 90.2/87.5 | 70.7/68.8 | 80.8 | 48.8 | 61.7 | 75.4 | 96.0 | 66.3 | 74.9 |
| LaCo@1/4-7B | 30.2 | 22.0 | 74.2 | 60.1 | 42.9/44.0 | 31.8/32.0 | 61.6 | 45.3 | 45.2 | 67.6 | 77.9 | 58.8 | 56.6 |
| LaCo@1-7B | 58.2 | 19.3 | 81.7 | 80.2 | 88.3/89.0 | 63.6/63.0 | 80.7 | 50.8 | 58.5 | 76.5 | 85.0 | 70.5 | 73.6 |

Table 11: Performance comparison between LaCo-7B and other MLLMs on multi-image benchmarks.

| Method | IEI | MI-VQA | NL-VR2 | Puzzle | Q-Bench | Spot-Diff | TR-VQA | VST | 3D-Chat | 3D-TD | ScanQA | ALFRED | nuScenes | BLINK | Mantis | MathVerse | MuirBench | SciVerse | Avg. |
|---|---|---|---|---|---|---|---|---|---|---|---|---|---|---|---|---|---|---|---|
| | | | in-domain multi-image | | | | | | in-domain multi-view | | | | | out-domain | | | | | |
| GPT-4V (OpenAI, 2023) | 11.0 | 52.0 | 88.8 | 17.1 | 76.5 | 12.5 | 54.5 | 10.9 | 31.2 | 35.4 | 32.6 | 10.3 | 63.7 | 51.1 | 62.7 | 60.3 | 62.3 | 66.9 | 44.4 |
| Mantis-7B (Jiang et al., 2024) | 11.2 | 52.5 | 87.4 | 25.7 | 69.9 | 17.6 | 45.2 | 12.5 | 2.6 | 14.7 | 16.1 | 14.0 | 46.2 | 46.4 | 59.5 | 27.2 | 36.1 | 29.3 | 34.1 |
| LLaVA-N-Iter-14B (Li et al., 2024d) | 24.5 | 95.0 | 91.1 | 59.9 | 76.7 | 40.5 | 78.6 | 33.3 | 70.6 | 52.2 | 34.5 | 62.0 | 76.7 | 52.1 | 66.4 | 33.4 | 40.7 | 32.7 | 56.7 |
| LLaVA-OV-7B (Li et al., 2024b) | 22.2 | 90.2 | 89.4 | 53.3 | 74.5 | 39.2 | 80.1 | 31.7 | 62.8 | 52.6 | 30.1 | 61.0 | 79.5 | 48.2 | 64.2 | 67.6 | 41.8 | 79.1 | 59.3 |
| LaCo@1/4-7B | 29.3 | 90.5 | 79.4 | 61.1 | 71.7 | 40.6 | 70.1 | 31.2 | 62.8 | 52.6 | 31.6 | 62.8 | 67.0 | 44.3 | 58.1 | 65.4 | 41.1 | 37.1 | 55.4 |
| LaCo@1-7B | 31.5 | 95.0 | 90.7 | 62.5 | 76.2 | 42.2 | 80.8 | 32.3 | 62.9 | 52.7 | 34.6 | 64.1 | 81.5 | 50.7 | 73.6 | 75.3 | 54.9 | 51.3 | 61.8 |

To investigate the performance of LaCo on larger-scale MLLMs, we construct the LaCo-7B multimodal model using AIMv2 as the vision encoder and Qwen2.5-7B as the language backbone. Specifically, LaCo@1/4-7B denotes applying LaCo to the 1/4 layers of AIMv2, while LaCo@1-7B represents applying LaCo subsequent to AIMv2. Additionally, we compare LaCo-7B against various open-source and closed-source multimodal models, with results presented in Tab.10, Tab.11, and Tab.12.

Tab.10 presents the comparison of LaCo-7B with other MLLMs on single-image benchmarks. Although LaCo@1/4-7B shows performance degradation compared to LLaVA-OV-7B, it achieves significant improvements in inference efficiency, with VET reduced by 78.3% and TPS increased by 40.1%, respectively. Notably, LaCo@1-7B outperforms GPT-4V across various benchmarks and surpasses GPT-4o on SeedBench and RealWorldQA. Compared to open-source models, LaCo@1-7B achieves comparable performance while reducing VET by 58.1% and improving TPS by 22.9% over LLaVA-OV-7B.

Tab.11 shows the comparison on multi-image benchmarks. LaCo@1/4-7B exceeds GPT-4V and Mantis-7B in terms of average metrics while performing comparably to other open-source MLLMs. LaCo@1-7B achieves the best performance among all models, demonstrating its strong capabilities in complex tasks such as multi-image reasoning, difference identification, and 3D environment understanding.

Table 12: Performance comparison between LaCo-7B and other MLLMs on video benchmarks.

| Method | EgoSchema | NextQA | PercepTest | S-bench | VideoMME | Avg. |
|---|---|---|---|---|---|---|
| | test | mc | val | video | wo/w-subs | |
| GPT-4V (OpenAI, 2023) | - | - | - | 60.5 | 59.9/63.3 | - |
| GPT-4o (OpenAI, 2024) | - | - | - | - | 71.9/77.2 | - |
| LLaVA-N-Video-32B (Zhang et al., 2024b) | 60.9 | 77.3 | 59.4 | - | 60.2/63.0 | - |
| LLaVA-OV-7B (Li et al., 2024b) | 60.1 | 79.4 | 57.1 | 56.9 | 58.2/61.5 | 62.7 |
| LaCo@1/4-7B | 41.9 | 64.9 | 47.9 | 40.7 | 46.9/55.1 | 49.3 |
| LaCo@1-7B | 52.7 | 77.1 | 57.3 | 57.3 | 58.8/62.0 | 61.0 |

Tab.12 presents the comparison on video benchmarks. LaCo@1-7B performs comparably to GPT-4V on SeedBench and VideoMME, though it still lags behind GPT-4o on VideoMME. Compared to open-source MLLMs, LaCo@1-7B shows overall comparable performance with advantages on SeedBench. The performance of LaCo@1/4-7B still has room for improvement, indicating that internal token compression within vision encoders warrants further investigation to achieve better results, particularly in larger model architectures.

