# OpenReview forum: "LaCo: Efficient Layer-wise Compression of Visual Tokens for Multimodal Large Language Models"
_ICLR.cc/2026/Conference — Submitted to ICLR 2026_

### Official Review · Reviewer_UqyW · 2025-10-24

**Soundness:** 2
**Presentation:** 3
**Contribution:** 2
**Rating:** 4
**Confidence:** 4

**Summary:**

This paper introduces Laco, a new method for visual token compression in MLLMs. The idea is to compress visual tokens within the intermediate layers of the vision encoder to mitigate token redundancy and enhance the efficiency of MLLM training and inference. Specifically, Laco employs pixel shuffle in intermediate vision encoder layers for compression, and a residual connection is incorporated to prevent the loss of information. The experimental results validate the superior performance of Laco compared to several baseline methods.

**Strengths:**

* A novel idea to incorporate layer-wise visual token compression into MLLMs
* Simple but effective method that can be easily applied to various MLLM architectures
* Extensive experiments to validate the effectiveness of Laco in multiple task settings and vision encoders

**Weaknesses:**

* The reported experiments are mainly based on MLLMs utilizing small 0.5B LLM backbones. Since the main contribution of Laco is the cost reduction in visual encoders, its overall impact on efficiency would be diminished when the LLM backbone scales up. While the paper provides results on 7B models in Appendix D, it lacks comparison of the overall computational efficiency of the MLLMs between Laco and the baseline methods. Besides, Loca@1/4-7B shows distinct performance degradation, which suggests that Laco may not be effective on larger models, with smaller efficiency benefits but large performance drops.
* The core contribution of Laco is the proposed layer-wise visual token compression. However, in the main experiments, baseline methods are also shifted to the intermediate layers, while Laco@1 is used for comparison against external compression. This experimental design is insufficient to properly validate the advantage of Laco.
* From my understanding, Laco can be seen as Pixel Shuffle + residual connection, and results of ‘Pixel-Shuffle’ can serve as ablations for the residual connection. So there can be ablations of the Pixel Shuffle component, e.g. applying the residual connection to other baseline methods.

**Questions:**

* There are layer-wise visual token compression methods for ViTs, e.g. ToMe[1]. How does Laco differentiate itself from simply incorporating such existing ViT techniques into the MLLMs?
* There is an interesting result that Loca@1/12 performs better than Loca@1/6. What would be the effect of applying Laco to compress the visual tokens before ViT (i.e., at the pixel level)?
* Have the authors tried to use Laco on multiple layers? (e.g. ToMe[1] merges tokens in each block of ViT)

[1]Bolya, Daniel, et al. "Token merging: Your vit but faster." arXiv preprint arXiv:2210.09461 (2022).

---

### Official Review · Reviewer_fb5f · 2025-10-28

**Soundness:** 2
**Presentation:** 2
**Contribution:** 2
**Rating:** 4
**Confidence:** 4

**Summary:**

This paper introduces LaCo, a framework for layer-wise compression of visual tokens within the vision encoders of multimodal large language models (MLLMs), in contrast to existing methods which typically compress tokens post-encoder. LaCo integrates a pixel-shuffle operation and a residual connection in a Patch Merge Layer (PML) and can be applied at different depths within the encoder. Extensive experiments on single-image, multi-image, and video benchmarks with multiple encoders (AIMv2, SigLIP, InternViT) are reported, demonstrating substantial improvements in efficiency (training time, inference throughput) with limited performance degradation compared to external compression baselines.

**Strengths:**

- The proposed method addresses an increasingly relevant challenge: mitigating the visual token bottleneck in MLLMs, which remains a significant computational and practical barrier in scaling such systems.
- The design of inserting the Patch Merge Layer (PML) at intermediate encoder depths and supplementing with a residual (space-to-channel + channel averaging) pathway is thoughtfully motivated and directly targets information loss from aggressive lossy compression.
- Thorough experimental evaluation: The paper presents comparisons across single-image, multi-image, and video benchmarks, including various vision encoders and multiple model architectures. Efficiency (both training and inference) is systematically quantified.

**Weaknesses:**

1. **Suboptimal Positioning vs. Most Recent Work**:
    - The Related Work section is incomplete with respect to several directly relevant recent advancements in token pruning and flexible token selection for MLLMs (e.g., LLaVA-Scissor, GlimpsePrune, TransPrune, FlexSelect, GreedyPrune, Beyond Attention/Similarity, explainability-driven compression, dynamic pruning in DynTok, Video Compression Commander, and SmolVLM). None of these are properly cited or contrasted, though they share substantial overlap in aims and evaluation targets with LaCo. This is a significant shortcoming—see "Potentially Missing Related Work" for specifics.
2. **Mathematical and Algorithmic Specification Insufficiency**:
    - The formulation of the residual connection in Section 3.3 (Equation for $RC(E_v^k, r)$) is only described at a high-level. The key details for the space-to-channel operation, channel averaging, and how gradients/learning interact across the shortcut and main path are not rigorously specified. For example, it is not clear from the equations how batch dimension is handled or whether normalization and residual addition are applied pre- or post-MLP. More formal clarity—ideally pseudocode or a concise algorithm box—is needed.
    - The loss/optimization details for training are not expressly given beyond “train PML in Stage 1.” Are additional regularization or auxiliary losses used to counteract information loss? Is there any adaptation or annealing of the compression ratio $r$? What is the effect of non-matching channels during the shortcut addition?
    - The notation in 3.2 (Token Compression) and 3.3 (Patch Merge Layer) is inconsistent—e.g., $E_v$, $E_v^k$, $\hat{E}_v$, and $\tilde{N}$ not clearly differentiated or indexed across encoder depths. More precise mapping and a notation table are needed.
3. **Empirical Evaluation Gaps and Limitations**:
    - Although the paper reports a wide range of benchmarks, some critical aspects are under-explored:
        - No in-depth error analysis or qualitative failure cases—what kind of visual information (e.g., text, fine structure, objects) is most lost at early compression? Figure 5 in Appendix B implies some drop in structured reasoning (e.g., DocVQA chart), but the paper only glosses over the challenge.
        - The efficiency gains are significant, but there is no report of GPU utilization, wall-clock time breakdown beyond tokens per second and encoding cost, or hardware/platform details. How does this compression interact with real-world memory bandwidth or latency bottlenecks?
        - For ablation (Table 7/8/9, Figure 4), the rationale for choosing $1/4$, $1/6$, $1/12$, and $1/2$ as candidate depths is empirical; is there a principled way to select these, or is it trial-and-error?
4. **Comparison Baseline Shortcomings**:
    - The implementation details of baselines (especially Pixel-Shuffle and TokenPacker) are not described; are these re-implementations faithful? Is code available? Variance across runs is not reported—are observed gains significant or within expected noise? Standard deviation/error bars are fully absent from all results tables.
    - No direct comparison to the very recent methods in “Potentially Missing Related Work”—for example, dynamic pruning and selection approaches, or explainability-based token compression.
5. **Clarity and Organization**:
    - Sections 3.3 and 4.1 lack diagrammatic clarity; equations and architecture overviews are mixed with narrative without a consolidated “Method Overview” diagram. Readers have to mentally map the architecture from text and piecemeal figures (e.g., compare Figure 2 and Figure 3), which reduces approachability for those less familiar with the pipeline.
    - References to equations use inconsistent notation (“Eq.2,” “Eq.1,” “Eq. 2,” sometimes not clear which displayed equation), which can cause confusion, especially when paired with narrative explanations.
    - Some claims in the abstract (“outperforms all existing methods in the intermediate layer scenario”) are overly broad given that not all recent methods have been compared, only classic/older ones.
6. **Limited Theoretical/Motivation Justification**:
    - While the design of layer-wise token compression is practically justified, there is no theoretical or information-theoretic argument—e.g., about redundancy distribution in early/mid/later encoder layers or about the impact of compression on key representation properties. An analysis (even empirical one, e.g., token-wise entropy or representational similarity before and after compression) would strengthen the work.
7. **Minor Issues**:
    - Typos and minor inconsistencies in section numbering (e.g., “Sec3.3” rather than “Sec 3.3”), references sometimes in inconsistent formats.
    - Appendix tables/figures referred to without in-text pointers specifying which results answer which research question.
    - “Compression ratio $r$” is treated as a hyperparameter, yet no tuning/dependence analysis is provided.
    - The claim on generalization, while evaluated over three encoders, is not extended to different LLM backbones or substantially different vision architectures.
In sum, the paper’s strengths are undercut by a lack of thorough positioning versus the most up-to-date related work, and a lack of algorithmic and theoretical depth in specifying precisely how and why the proposed solution excels.

**Questions:**

Please see the weakness

---

### Official Review · Reviewer_oiGJ · 2025-10-31

**Soundness:** 2
**Presentation:** 2
**Contribution:** 2
**Rating:** 2
**Confidence:** 5

**Summary:**

This paper focuses on visual token compression for Multi-modal Large Language Models. Instead of  post-encoder operations, Layer-wise Visual Token Compression (LaCo) is proposed to  perform within the vision encoder’s intermediate layers, through a layer-wise pixel-shuffle and a residual learning path. Experiments are carried out following LLaVA-OneVision.

**Strengths:**

[+] The manuscript is well written.

[+] Experiments are conducted in a series of MLLM benchmarks, following the standard  pipeline of LLaVA-OneVision.

**Weaknesses:**

[-] Novelty. Many existing works have explored token compression in the visual encoder of MLLM, but these have been largely overlooked in the related work. Overall, the idea presented in this paper is trivial and contribute minor to the community.

Token merging: Your vit but faster.

Spvit: Enabling faster vision transformers via soft token pruning.

Not all patches are what you need: Expediting vision transformers via token reorganizations.

FOLDER: Accelerating Multi-modal Large Language Models with Enhanced Performance.


[-] Across Table 1-6, the comparisons against the baseline focus solely on performance, without controlling for compression efficiency or, more importantly, metrics like throughput and GFLOPs. This one-sided emphasis on performance without demonstrating efficiency is problematic and makes the paper's contribution unclear.

[-] Token compression in the visual layer is fatal for interactive image-text understanding and QA. For example, if the first text query asks about the main subject of the image, many visual background tokens get compressed; if the second query then asks about the image’s background, the model will be unable to handle it.

**Questions:**

[-] Architecture generalization. Is the conclusion of this paper effective and equally effective for models of different scales? How to ensure that the hyperparameters involved in the method, such as compression layer index and number of compression layers, are optimal?

---

### Official Review · Reviewer_wPB8 · 2025-10-31

**Soundness:** 2
**Presentation:** 3
**Contribution:** 2
**Rating:** 4
**Confidence:** 3

**Summary:**

The paper proposes LaCo, a method for compressing visual tokens in Multimodal Large Language Models (MLLMs). The authors identify that most existing compression methods operate after the vision encoder (externally), which is computationally inefficient as the full-length token sequence must be processed by the entire encoder. LaCo performs the compression within the vision encoder's intermediate layers. LaCo is composed of a layer-wise pixel-shuffle mechanism to merge adjacent tokens and a residual learning architecture with a non-parametric shortcut to mitigate information loss during this merging process.

**Strengths:**

1. Clear presentation. The paper demonstrates its method and motivation clearly.
2. Considerable experimental results: When comparing LaCo to other compression methods (Pixel-Shuffle, LDPv2, TokenPacker) all placed at an intermediate layer (the 1/4 layer), LaCo (53.6 Avg) dramatically outperforms the others (all ~36 Avg).
3. Good ablation: The paper provides a clear and useful ablation study on the effect of the compression layer's depth (e.g., 1/12, 1/6, 1/4, 1/2) . This analysis correctly identifies the trade-off: compressing earlier maximizes efficiency gains but hurts performance the most, while compressing later (e.g., 1/2) offers a better balance.

**Weaknesses:**

1. Core claim contradiction: The paper's central argument is that internal compression is superior. However, the exp data for the 0.5B models (Tables 4, 5) show this is not the case for performance. When comparing internal (LaCo@1/4) vs. external (LaCo@1) compression, external compression outperforms internal ones in a lot of cases.
2. Short of experiments: The paper only conducts experiments on 0.5B models. Without the evidence that their claims can extend to larger models.
3. Misleading baseline comparisons: The main win in Table 1 is achieved by comparing LaCo@1/4 against other methods also placed at the 1/4 layer. It only proves that LaCo's residual connection is necessary for any internal compression to work, but it doesn't prove LaCo@1/4 is better than, for example, the external versions of those baselines.

**Questions:**

See Weakness

---

### Meta-Review · Area_Chair_PA3v · 2025-12-03

**Summary:**

This paper introduces LaCo, a method for compressing visual tokens within a vision encoder to improve MLLM efficiency. Reviewers praised its clear presentation and thorough experiments but raised major concerns. The claimed superiority of internal compression is contradicted by the paper's own results which favor external compression on 0.5B models. Validation is limited to small models, with a 7B model showing performance degradation. Comparisons are misleading, as advantages are shown against internally adapted baselines rather than native external ones. The related work section omits key recent studies. There is a lack of algorithmic detail, theoretical justification, and analysis of information loss. The authors' rebuttal provided clarifications but failed to resolve the core contradictions, limited model scale testing, incomplete baseline comparisons, and literature gaps, leaving the paper's contributions unsubstantiated.

**Reviewer Concerns:**

The rebuttal provided some efficiency data for larger models and clarifications on baseline implementations, partially addressing concerns about efficiency reporting and experimental details. However, the core concerns regarding the contradictory evidence for the main claim, the lack of convincing results at scale, the insufficient comparison to standard external baselines, and the omission of key related work remain fully outstanding.

**Reviewer Scores:**

Given that the rebuttal did not resolve the fundamental methodological and empirical shortcomings highlighted in the discussion, it is unlikely any reviewer would have raised their score. The reviewers who initially indicated a marginal score (4) would likely maintain or potentially lower it due to the unaddressed core issues, while the reviewer who suggested reject (2) would find their concerns reinforced.

---

### Decision · Program_Chairs · 2026-01-26

Reject